# FSTEGA: FOURIER NEURAL OPERATORS FOR PRINTER-PROOF STEGANOGRAPHY

## ABSTRACT

Hiding and extracting a message in printed images is challenging when considering the trade-off between accuracy in recovering data and the perceptual quality of the generated images. It is especially a big issue to hide data in such a way that it is almost invisibly embedded. In this paper, we propose a method based on Fourier Neural Operator to embed bitstrings in images. The method is able to learn critical frequencies from the image and the message to improve the decoding process of the hidden data. In order to enhance the information recovery capabilities of the printed image we create an improved noise simulation process and create a decoder composed by several convolutional layers combined with a vision transformer to obtain a decoding method more robust to the noise introduced in the encoding image when it is printed and acquired by an optical sensor. Experimental evaluations demonstrate the ability to properly recover the message encoded in wild printed pictures with and accuracy of 100% (with 2 different image sizes) acquired with several lightning and perspective conditions.

## 1 INTRODUCTION

Nowadays, we have widespread applications that use QR codes and similar technologies to share information such as URLs, purchase goods, or even make bank transactions and validate documents. Usually, those strategies use graphic codes with information embedded where the data can be recovered using regular cameras, like the ones present in common smartphones. Due to security concerns and also aesthetic reasons, it is important to seek alternative strategies that eliminate or reduce the presence of visible codes such as bars, fiducial markers, or QR codes.

Steganography is the process of hiding secret information (e.g. text, image, or video) inside an inconspicuous cover medium, which for our purposes is an image. The secret information should be invisible to the human eyes, therefore only the sender and the intended receiver should realize the existence of secret information in the cover image. Watermarking is a technique similar to steganography which aims to embed a message in a cover image to verify ownership, meaning that the message must be robust to noise and tampering from third parties (Mandal et al., 2022). Several papers have been proposed the use of deep learning techniques for watermarking and steganography (Baluja, 2017; Tancik et al., 2020). The main idea is to assign a bit array to a string and then convert it to an image pattern and insert it into an image. We focus specifically on printer-proof steganograpgy issues, which require robust encoding and decoding algorithms. The encoded image must successfully hide the secret information, and the decoding method needs to be robust to noise that occurs during printing and scanning processes and to other physical distorsions. The state-of-the-art work in this area, StegaStamp (Tancik et al., 2020), presents the first steganography printer-proof model, based on deep convolutional neural networks. To handle the distortions that can result from the printing process, StegaStamp simulates several image corruptions between the encoder and the decoder. In order to preserve good perceptual image quality, the authors propose a new loss function which is a weighted sum of different components like $L2$ distance between images, the LPIPS loss (Zhang et al., 2018) and cross-entropy. However, it has some limitations, namely 1) the residual added by the encoder network is sometimes very perceptible in low-frequency regions of the image, 2) it has difficulty in decoding images with sizes smaller than 15 cm width, and 3) the method was evaluated using controlled light conditions.

In this paper, we propose a framework for robust printer-proof watermarking, based on deep learning models. Our architecture for encoding and decoding uses Fourier Neural Operators instead of regular convolutions. The main objective is to learn sensitive image frequencies aiming to produce robust patterns against printing noise. Additionally, we developed a robust noise simulation pipeline based on physics color augmentation to represent the changes in different light conditions when capturing images with regular cameras.

## 2 RELATED WORKS

The aim of steganography is covert communication, and watermarking is commonly used to assign ownership. Although steganography and invisible watermarking have different goals, both techniques share a similar process, successfully hiding and extracting secret information. Since these two techniques include the same steps, namely, secret information, the encoding algorithm, and the decoding algorithm, they can be comparable. Our work focuses on invisible watermarking.

### 2.1 WATERMARKING

Zhu *et al.* proposed HiDDeN ((Hiding Data With Deep Networks) (Zhu et al., 2018) which is, to the best of our knowledge, the first end-to-end neural network approach to embed a watermark in a cover image. The authors propose an encoder-decoder architecture to combine a cover image with a message and retrieve the message from an encoded image. Since robustness is a central motif for watermarking, the authors propose to increase it by inserting a noise layer that distorts the encoded image before feeding it to the decoder, thus simulating loss of quality in the images during transmission and capture. They also propose the use of an adversarial module to learn whether an image has an encoded message or not (a binary classifier). This way, it forces the encoder to hide the message in an imperceptible and robust manner, while forcing the decoder to extract the message from an encoded image even under challenging distortions to the image to be decoded. While their work presented a significant improvement of the state-of-the-art at the time, the efforts were focused only on digital images.

Bui et al. (2023) proposed RoSteALS, which is a watermarking method that injects a secret message directly into the latent code of a frozen autoencoder, enabling robust watermarking with limited training. The authors state that the model is robust against image perturbations; however, they do not compare the results with real printed images.

We propose an approach focusing on printed images, namely images printed on paper and other substrates, thus going a step further compared to HiDDeN and RoSteALS.

### 2.2 STEGANOGRAPHY MODELS

Using deep learning, image steganography has emerged in the last years as one of the most accurate techniques to hide messages in images. Several works have been recently proposed (Baluja, 2017; Lu et al., 2021; Tancik et al., 2020). In this context, HiDDeN (Zhu et al., 2018) may also be used for steganography and presents the same limitations as stated before.

Lu et al. (2021) introduced an invertible neural network model to solve image steganography and recovery data problems. They focus on a bijective transformation model that uses a single network to hide and reveal data efficiently in images. However, similarly to HiDDeN, that method was not tested for printed images. StegaStamp, proposed by Tancik *et al.* (Tancik et al., 2020), claims to be the first steganography model capable of decoding data from printed images. The authors show robust results in decoding data under physical transmission by developing novel strategies to add noise in the training process, printer noise simulation, and distortion for the training dataset. However, their work has two main limitation: (i) it is limited to square images surrounded by a fairly large "dead" area; (ii) the authors mention that their approach adds residual noise to uniform regions of the container image, i.e., regions with low frequency, thus limiting the desired effect of steganography.

We propose to employ Fourier Operators Li et al. (2020) to encode the message predominantly on regions of medium to high frequencies of the cover image. Additional details are provided in Section 3.1. This approach may partially solve the problem of undue noise introduction in uniform

regions of the cover image, since the insertion of artifacts will be restricted to regions with richer information density, i.e., higher frequencies, thus have higher chance of passing undetected by an unaware observer.

## 3 METHOD

### 3.1 ENCODER

Our encoder is based on the U-Net architecture (Du et al., 2020; Xu et al.; Graham et al., 2021). However, instead of using regular convolutions, we apply Fourier Neural Operator in each hidden layer. Also, on the bottleneck part, we add a Vision Transformer(Han et al., 2022; Wang et al., 2021; Dosovitskiy et al., 2020), which enables local-to-global reasoning. Similarly to StegaStamp, as input, we receive a six-channel 256 × 256-pixel tensor and output a three-channel RGB residual image. The input message, i.e, a string, is converted into a 100-bit binary array and processed by a fully-connected layer to create a 256 × 256 × 3 tensor. Then, this tensor is concatenated to the input image to form the input of the U-net-based architecture. Since our model uses Neural Operators instead of only regular convolutions, the first layer is used to lift the input data to a higher-dimensional channel space. To do that we apply a 2D convolutional Layer. To both the downsampling and upsampling parts of the U-Net, we apply 5 layers composed of 2D convolutions and Neural Operators. Also, at the upsampling part, we combined the encoded features with data from skip connections. This second part of the U-Net upsamples the encoded features, which are then combined with the high-resolution details by recovering localized spatial information. The last layer of our encoder model is a pointwise convolution, used to project back the data to the target dimension and to produce the residual image. Figure 1 shows the architecture of our encoder.

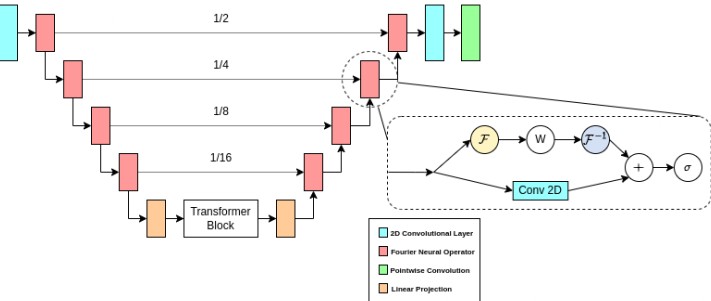

Figure 1: Block diagram of the encoder architecture. Each Fourier Neural Operator is composed of 2 branches and the first one is composed of a regular convolution. The second branch starts by performing a Fourier transform, followed by a linear operation W in frequency space, and finally an inverse Fourier transform to get back to the original image domain is performed. After that we sum the outputs from both branches and apply a GELU activation function represented by $\sigma$. Our network structure is based on the U-Net, however, the first layer lifts the data to a higher-dimensional channel space, and the last 2D convolution project back to the target dimension. In the end, we use a pointwise convolution to reduce the number of channels to be the same as the RGB image.

#### 3.1.1 FOURIER NEURAL OPERATOR

Our Encoder consists of a U-Net-based architecture that contains several spatial Fourier transform modules instead of regular convolutions. This is our main building block to encode messages into images, and it is inspired by the Fourier Neural Operator (FNO) (Li et al., 2020). The idea is to build an image-to-image mapping that can be used to minimize colored artifacts (distortions) while maintaining the robustness of our model to encode data into images. Let's consider that $\mathcal{X}_l \in \mathbb{R}^{w,h,c}$ defines an input tensor where $w, h$ and $c$ are its width, height and the number of channels, for a layer $l$, i.e., an image as input for our encoder and $\mathcal{X}_{l+1}$ its output. The Neural Fourier Operator can be written in the form described in Equation 1:

$$\mathcal{X}_{l+1} = \sigma(C_l(\mathcal{X}) + \kappa_l(\mathcal{X})) \tag{1}$$

where $C_i$ is a linear mapping representing a convolution operation, $\kappa$ is the Fourier operator over $\mathcal{X}$ and $\sigma$ is the activation function. The Fourier operator can be described as the equation 2 Li et al. (2020):

$$\kappa_l(\mathcal{X}) = \mathcal{F}^{-1}(W_l \cdot (\mathcal{F}(\mathcal{X})))$$ (2)

where $\mathcal{F}(\mathcal{X})$ is the 2D Discrete Fourier Transform of input tensor $\mathcal{X}$; $W_l$ is the trainable weights of a layer $l$ and $\mathcal{F}^{-1}$ is the inverse 2D Discrete Fourier transform. Note that, as described by Li et al.(Li et al., 2020), we perform a convolution in frequency space and we are interested in performing a linear transform on the lower Fourier modes and filtering out the higher modes. Since the operation convolves $\mathcal{X}$ with a function that only has $k_{max}$ Fourier modes, we truncate higher frequency signals with a window size of $k$, i.e., the linear transform $W_l \cdot (\mathcal{F}(\mathcal{X}))$ can be rewritten as equation 3Chen et al. (2022):

$$\sum_{i=1}^{c} W_{i,j,u,v} \cdot F_{i,u,v}, \ u, v = 0, \cdots, \pm k, \ j = 1, \cdots, c$$ (3)

where $F_{i,u,v} \in \mathbb{C}^{c,2k+1,2k+1}$ is the truncated frequency domain representation of $\mathcal{X}$ after performing the 2D Discrete Fourier Transform $\mathcal{F}(\mathcal{X})$, $W \in \mathbb{R}^{c,c,2k+1,2k+1}$ is the layer weights and $c$ is the number of input channels.

By using Fourier Neural Operators, our model is capable of learning a set of low frequencies from images and from the message to be encoded in such a way that it can improve the robustness against printer noise. In other works, such as StegaStamp, the residual added by the encoder network is sometimes perceptible in low-frequency regions of the image. As we can see in Figure 2, our strategy also tries mitigates this problem.

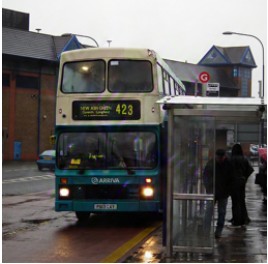 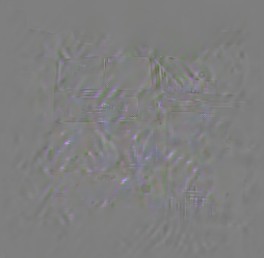

(a) encoded image          (b) residual

Figure 2: Message encoded using FStega. On the left, the encoded image. The residual is shown on the right. Notice that most of the residuals are encoded in medium-high frequency regions of the image, notably in the center, near the bus.

## 3.2 DECODER

Our decoder is a neural network trained with encoded images aiming at recovering the hidden message. It is composed of 5 layers of 2D Convolutions, combined with 2 Fourier Neural Operators after the second and fourth layers, followed by a Transformer layer and 2 Dense layers. During training, our decoder receives noisy images, which have passed through a noise simulation module. Those images were processed to simulate real-world scenarios where a printer-proof method is required. In such scenarios, the encoded images are first printed and then captured by an optical sensor (e.g. smartphone camera). Each one of the steps adds its own type of noise, degrading the quality of the image to be decoded. Additionally, our decoder was trained using the binary cross-entropy loss.

## 4 NOISE SIMULATION FOR PRINTER-PROOF ROBUSTNESS

Our model needs to be robust to recover messages embedded in printed images. To do so, during the training, we combine and improve the ideas presented by StegaStamp (Tancik et al., 2020), HiDDeN (Zhu et al., 2018) and Deep ChArUco (Hu et al., 2019). We simulate perspective warping, motion

blur, printing, light conditions, color manipulation, and camera noise. For perspective warping, we apply the same idea as in StegaStamp. A random homography is peformed to simulate the effect of a camera, by perturbing the four corner locations of the image uniformly within a fixed range of ± 25 pixels. To simulate motion blur, we sample a random angle between 90 and 180 degrees and generate a 5 × 5 pixels size blur kernel. Our improvements to noise simulation consider other physical behaviors. We develop a different physical display-imaging pipeline by using Planckian color jittering (Zini et al., 2022), posterization, and plasma fractals (Fournier et al., 1982). These processes are explained in the following section.

## 4.1 Color Manipulation

We apply a random transformation to the brightness, contrast, saturation, and hue value of images. We also use fractals as a means to manipulate contrast and randomly adjust hue, brightness, and saturation. We employ plasma fractals for adapting global image augmentation transformations into continuous local transforms. For the contrast manipulation, we use the strategy from Nicolaou et al. (Nicolaou et al., 2022), where the Convolutional Diamond Square algorithm is applied with the roughness parameter varying from 0.3 to 0.5. For the Hue shift, we randomly shift the RGB channels with values sampled uniformly from an interval of -0.15 to 0.15. Brightness is defined as an additive operation directly to the raw pixel where the values are shifted according to a factor sampled between 0.3 and 0.5.

## 4.2 Light simulation

We use a physics-based color augmentation, called Planckian jitter, which creates realistic variations in chromaticity. This algorithm starts by calculating a sample for a new illuminant spectrum $\sigma_T(\lambda)$ from the distribution of a black body radiator, where it can be calculated at temperature $T$ using Planck's Law (Zini et al., 2022; Andrews, 2010):

$$\sigma_T(\lambda) = \frac{2\pi hc^2}{\lambda^5(e^{\frac{hc}{kT\lambda}} - 1)} \mathbf{W/m^3},$$

(4)

where $c = 3 \times 10^8$ m/s is the light speed, $h = 6.63 \times 10^{-34}$ Js is the Planck constant, $k = 1.380662 \times 10^{-23}$ J/K is the Boltzmann's constant and $T$ is randomly sampled in the interval between 3000K and 15000K, which can be encountered in real-world environments. Also, the $\lambda$ component is randomly sampled between the interval of 400nm and 700nm. After the generated spectrum is converted to its sRGB representation and we created a jittered image combining the input image with the sRGB representation of the spectrum, through a simple channel-wise scalar multiplication.

## 4.3 Printing Simulation and Camera Noise

To simulate printed images, we apply "posterization" over images. We manipulate the color histogram of images by changing their 8-bit representation to a 7-bit representation for each color channel. As in the proposed technique by Chao et al. (Chao et al., 2021), we take an encoded image as input and posterize it by partitioning it into regions with smooth boundaries where possible while preserving essential details, where region colors are a close visual match to the original image.

Several works explore the camera systems and propose algorithms to simulate them. In our work, we chose to simulate the photon shot noise(Beenakker & Patra, 1999) by just using Poisson noise. Our image is $256 \times 256$ pixels, and we set a mean incident photon flux of 500 photons on each pixel.

## 5 Implementation and training details

## 5.1 Training Datasets

In our experiments, we train our model with 2 Datasets: Common Objects in Context (COCO) (Lin et al., 2014) and CelebA (Zhang et al., 2020) dataset. The images were rescaled to a $256 \times 256$ resolution and combined with binary messages converted to a tensor with 3 channels. For COCO dataset, we use 120 000 images for training and 40 000 for testing. With CelebA we use 160 000 images for training and 20 000 for testing.

## 5.2 LOSS FUNCTION AND OPTIMIZATION

Our loss function is composed of three components, i.e., we train our model to minimize the Focal Frequency Loss between the encoded image and the original, the LPIPS perceptual loss (Zhang et al., 2018), and cross-entropy loss for the message. The Focal Frequency Loss is a frequency-level objective function that helps to refine the generated frequencies from encoded images to improve output quality. The Focal Frequency loss (FL) is defined by the frequency distance between the images and the weight for the spatial frequency, as in the equation 5 (Jiang et al., 2021)

$$FL = \frac{1}{MN} \sum_{u=0}^{M-1} \sum_{v=0}^{N-1} w(u,v) |F_r(u,v) - F_f(u,v)|^2 \tag{5}$$

where $(u,v)$ represents the coordinate of a spatial frequency on the frequency spectrum, $F(u,v)$ is the value of the component $(u,v)$ for a 2D discrete Fourier transform, M and N represent the width and height of an image, and $w$ is the weight for the spatial frequency at $(u,v)$.

Our main loss function is, thus, defined by a weighted sum of three loss functions and as in the Equation 6.

$$Loss = \lambda L_{lpips} + \omega L_M + \gamma FL \tag{6}$$

namely, the LPIPS perceptual loss, $L_M$ as the cross-entropy for the binary messages, and $FL$ as the focal frequency loss. In our experiments, we defined the weights as $\lambda = 1.5$, $\omega = 4.0$, $\gamma = 1.5$. We use the AdaBelief optimizer(Zhuang et al., 2020) for our training process. This optimizer achieves a more stable training and fast convergence then the adaptive methods while having a good generalization as in Stochastic Gradient Descent(Sutskever et al., 2013). We use a learning rate of $1 \times 10^{-4}$ and $\epsilon$ of $1 \times 10^{-15}$.

## 6 EXPERIMENTS AND EVALUATION

### 6.1 ABLATION STUDY FOR ENCODING AND DECODING

We performed training experiments that excluded the FNO structure and carefully analyzed the resulting outcomes. During this process, we utilize the COCO dataset. To compare the models with and without the FNO block, we employ the test set and incorporate our complete pipeline for noise simulation. The model that includes FNO layers achieves an accuracy of 97.5% in recovering the bit-strings, whereas the accuracy drops to 87% when using a regular U-Net for the encoder and a decoder without FNO. When considering the parameter count of our model compared to the state-of-the-art StegaStamp, we can see a slight increase. This difference arises from incorporating the Fourier Neural Operator and the Vision Transformer in our encoder architecture. However, our experiment demonstrated that this architectural choice enhances the control over encoded images by considering the frequency components present in both the original image and encoded data, resulting in improved accuracy. To address the parameter count issue, we mitigated it by compressing these FNO layers using Tucker decompositions. With this factorized version, our encoder achieves only 5% more parameters compared to Stegastamp.

We test how our noise simulation process affects our model. In the first experiment, we considered 4 versions of our model where each version was trained considering subsets of the noise simulation pipeline. The first model was trained to consider only changes in perspective and color jittering. The second one receives only perturbations considering lighting changes. The third considers only changes in printing simulation and the last one considers the whole simulation pipeline. For testing, we randomly select 2000 images from the test set of the COCO dataset and we encode messages with 100 bits of information. We analyze how each perturbation subset affects each model, in a similar process as in StegaStamp (Tancik et al., 2020). As we can see in Table 1, the model which was trained only with the "posterization" noise is robust enough when processing but has difficulty decoding warped images. The first and second models are also robust against the posterization process and, as expected, the model trained with the entire pipeline has the best performance.

### 6.2 BENCHMARK

We performed a set of experiments by printing and capturing encoded images to demonstrate the robustness of our technique. We encoded and printed 100 images randomly selected from the COCO

| Models | Posterization | Warp simulation + color jittering | Light simulation |
|---|---|---|---|
| warp simulation + color Jittering | 95.63% | 96.31% | 94.92% |
| Light | 94.05% | 85.46% | 94.74% |
| Posterization | 95.49% | 75.05% | 86.13% |
| Complete Pipeline | 96.22% | 97.56% | 94.74% |

Table 1: Accuracy of the trained models to recover the bit-strings using different transformations for training and testing. The rows show the transformations used for training, while the columns present the transformations used for testing.

dataset and embedded 100-bit messages into them. Those messages are hash codes with eight characters related to the filename of each picture. Those hashes are then converted into bit strings and processed using the LDPC algorithm, a linear error-correcting code (Gallager, 1962). Unlike in work (Tancik et al., 2020), we do not conduct experiments in a controlled environment with fixed lighting. We perform real-time detection using a cellphone camera in an uncontrolled environment and with ambient light.

The printed images are equaled positioned relatively to the camera and the surrounding illumination sources, for consistency, and captured by a camera from a Samsung Galaxy S22 Ultra. The resulting photographs are cropped and rectified. The cropping procedure is performed by classic image processing methods. Each printed image was augmented by a magenta border with 5 pixels on each side. After capturing, each image is converted to grayscale and binarized with a threshold value of 160. This threshold was found empirically, with the goal of isolating the interior of the magenta border from the rest of the paper sheet. After that, we apply a binary hole-filling algorithm and detect bounding boxes of all regions with an area larger than 20.000 pixels, thus diminishing the chances of spurious detection. Afterward, all regions with an aspect ratio larger than 0.82 are selected for decoding. Again, this value was found empirically. The complete procedure is presented in the Supplementary Material.

We focused on testing the decoding capabilities of FStega and StegaStamp since the latter is the state-of-art of printer-proof general steganography model. We printed the test images varying their sizes with 10x10cm and 15x15cm. Ultimately, we tested the decoding process over 100 printed images of each size, totalling 200 printed images. We captured ten photos of each printed image, where, in total, we have 1000 samples for each image size (totalling 2000 samples). Figure 3 shows 2 samples of printed images with colored borders for detection.

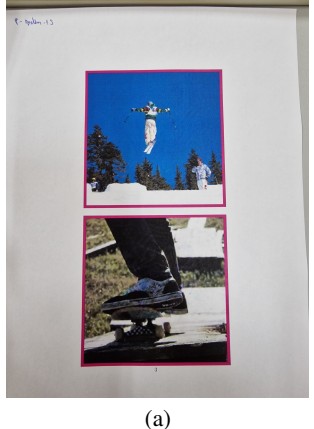 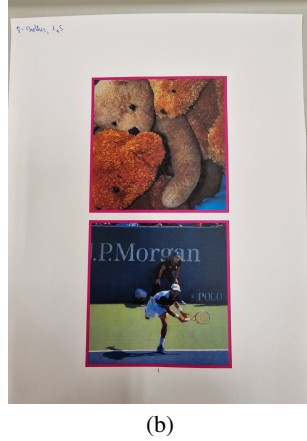

(a)             (b)

Figure 3: Printed images with size 10x10 cm in A4 paper sheets. Note that, although we can decode the messages properly, some artifacts are still visible.

| Model | 10x10 cm | 15x15 cm |
|---|---|---|
| StegaStamp | 99% | 100% |
| FStega | 100% | 100% |

Table 2: Percentage of printed photos successfully decoded by StegaStamp and FStega.

The percentage of successfully decoded images for both methods is presented in Table 2. FStega obtained a decoding accuracy of 100%, meaning that all printed images were decoded considering the sizes of 10x10 cm and 15x15 cm. On the other hand, StegaStamp also had stellar results for 15x15 cm, although it missed two images on 10x10 cm. Furthermore, StegaStamp encodes 100 bits of information on a $400 \times 400$ image, while the herein proposed method encodes the same information on a $256 \times 256$ image; thus, we have more information density. For a fair comparison, we use the bits-per-megapixel metric, which normalizes the message length and image sizes between different methods. For the channel capacity, we have 1402 bits-per-megapixel, while the state-of-the-art StegaStamp produces 571 bits-per-megapixel.

We also compare our model against other approaches. We compare our model against StegaStamp(Tancik et al., 2020), Hidden(Zhu et al., 2018),Light field messaging with deep photographic steganography(Wengrowski & Dana, 2019) and RoSteALS Bui et al. (2023). It is worth emphasizing that among the other evaluated methods, StegaStamp stands out as the sole approach explicitly developed for the primary purpose of decoding printed images. The tests were performed in a digital environment using the pipeline that simulates the printing process. Our model outperforms previous works in terms of accuracy. Hidden achieves an accuracy of 65%, while LFM achieves 63% accuracy. Both StegaStamp and RoSteALS achieved a consistent accuracy of 99%. In comparison, our model achieves a remarkable 100% accuracy. We compare in Table 3 the image quality of the FStega method and the other methods using as quality metrics the peak signal noise ratio (PSNR), the learned perceptual similarity metric (LPIPS), and the structural similarity (SSIM). The results show that the FStega method presented better SSIM and PSNR values, and RoSteALS achieved better LPIPS value.

| Model | SSIM ↑ | PNSR ↑ | LPIPS ↓ |
|---|---|---|---|
| StegaStamp | 0.89 | 28.46 | 0.029 |
| DeepStega | 0.92 | 24.60 | 0.260 |
| HiDDeN | 0.77 | 24.50 | 0.201 |
| RoSteALS | 0.93 | 30.36 | **0.027** |
| FStega | **0.94** | **30.51** | 0.030 |

Table 3: Image quality metrics across different approaches. Note that for SSIM and PSNR, higher values are better. For LPIPS, the opposite is true.

### 6.3 LIMITATIONS

One of the main limitations of our model relates to resizing encoded images. Our model receives as input images with the size of $256 \times 256$ pixels. If an image has smaller dimensions than the input requirements, we need to rscale-up the image and, after the encoding process, resize the output to the original dimension. By downsampling and upscaling the image again to decode, part of the hidden message may be corrupted. We test this hypothesis with 1000 images from the CelebA dataset, where by doing this process, our accuracy drops from 98% to 45%. We've partially circumvented this limitation by upscaling the output of our model to the container image size using a Nearest Neighbors interpolation. This residual is then added to the container to produce the encoded image. This procedure results in better-quality images without loss of decoding capabilities. However, more experiments may be performed to assess this process. Furthermore, comparing the images produced by FStega and StegaStamp, there is a concentration of artifacts near the center of the image for FStega. This feature adds robustness to our method, and is not noticeable for images with color variations near the center (see Fig. 4); however, for images of homogeneous colors (such as the one in Fig. 5), the changes produced by the encoder are noticeable. It should be noticed that this is also a limitation of StegaStamp (see Fig. 5b). This indicates that the proposed printer-proof

steganography method can be further improved, namely, making the residue less noticeable without severely impacting the decoding capabilities of the model.

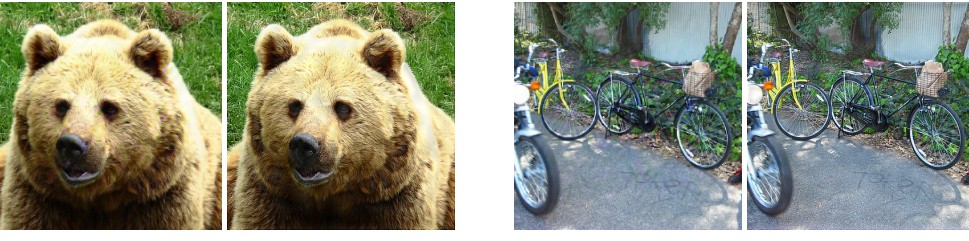

(a) FStega (left) StegaStamp (right)  (b) FStega (left) StegaStamp (right)

Figure 4: Visual comparison between the images produced by FStega and StegaStamp
.

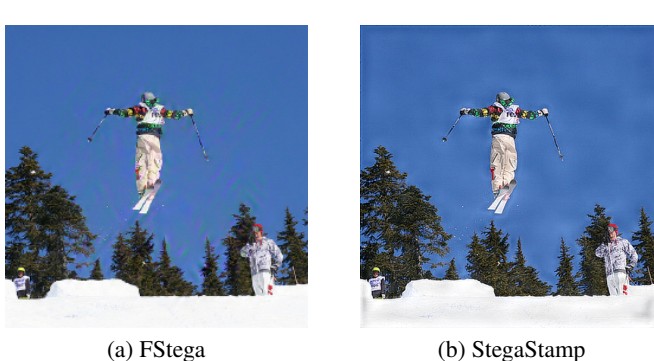

(a) FStega  (b) StegaStamp

Figure 5: Visual comparison between images with large homogeneous images encoded with FStega (Fig. 5a) and StegaStamp (Fig. 5b).

## 7 CONCLUSIONS

We presented a deep end-to-end learning architecture for printer-proof steganography. We encode messages with a string corresponding to a 100-size-bit array, including correction error codes. Our method is competitive considering the current state-of-the-art. We improve the image perturbation process of the state-of-the-art to simulate realistic conditions. However, for printer-proof approaches, we prove that creating images with small level of distortion is almost impossible, considering data artifacts, such as small color changes and low-frequency distortions. In contrast with the previous works, we presented a printer-proof steganography solution that proved to be robust to real-world scenarios, without controlled lighting systems. Our model is similar to StegaStamp in accuracy; however, we have almost double the capacity for message encoding considering the bits-per-megapixel measure. In future work, we aim to explore new possibilities for reducing image artifacts by including a better evaluation of the frequency range where we can hide the message while minimizing image perturbations.

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

## A  APPENDIX

### A.1  ARCHITECTURE DETAILS

Here we describe in detail the Network architectures for our encoder and decoder as shown in Table 4; our encoder model comprises 22 layers of 2D convolutions, Fourier Neural Operators(Chen et al., 2022; Li et al., 2020) and Upsampling layers, and a Vision Transformer(Xu et al., 2021). Our decoder comprises ten layers, with five 2D convolution layers, two Fourier Neural Operators, one Vision Transformer, and two fully connected layers. We show our decoder architecture in Table 5 and in Figure 6 as well.

### A.2  NETWORK RECOVERY CAPACITY

To measure how much information we can encode in images considering our network, for evaluation, we use the same metric as in StegaStamp (Tancik et al., 2020). We treat the mean bit recovery accuracy as the crossover probability $p$ in a binary symmetric channel, so we also can use information theory to calculate the channel capacity as in the equation(Tancik et al., 2020):

$$C(p) = 1 - (-p \times log_2(p) - (1 - p) \times log_2(1 - p)) \tag{7}$$

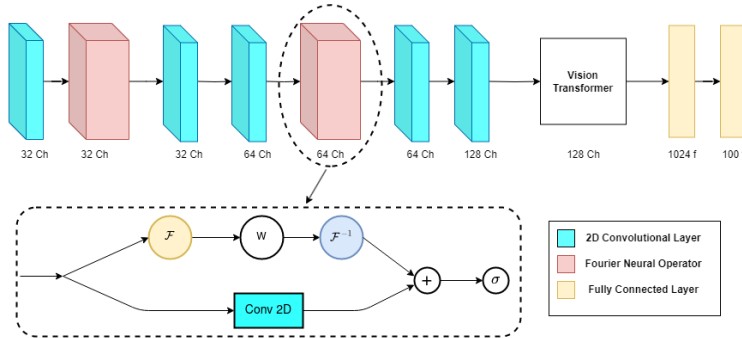

Figure 6: Block diagram of the decoder architecture. The decoder architecture consists of five 2D convolutional layers, two Fourier Neural Operators (applied after the second and fourth layers), a Transformer layer, and two Dense layers.

Table 4: We identify convolutional layers with the prefix "conv", the visual transformer as "Vit", Fourier Operators as FOP, and Max Pooling operations as MaxPool. A GELU is applied after each regular layer and each "conv + Fop" operation.

| Layer | kernel | ch | input | modes |
|---|---|---|---|---|
| Inputs | | 6 | image + code | |
| Conv1 | 3 | 6/32 | inputs | |
| Conv2 | 3 | 32/32 | conv1 | |
| Fop1 | | 32/32 | conv1 | 32 |
| Conv3 | 3 | 32/64 | MaxPool(conv2+Fop1) | |
| Fop2 | | 32/64 | MaxPool(conv2+Fop1) | 16 |
| Conv4 | 3 | 64/128 | MaxPool(conv3+Fop2) | |
| Fop3 | | 64/128 | MaxPool(conv3+Fop2) | 8 |
| Conv5 | 3 | 128/256 | MaxPool(conv4+Fop3) | |
| Fop4 | | 128/256 | MaxPool(conv4+Fop3) | 8 |
| Vit | | 256/256 | MaxPool(conv5+Fop4) | |
| up6 | 2 | 256/128 | UpSample(Vit) | |
| Conv6 | 3 | 256/128 | MaxPool(conv4+Fop3)+up6 | |
| Fop6 | | 256/128 | MaxPool(conv4+Fop3)+up6 | 8 |
| up7 | 2 | 128/64 | UpSample(conv6+Fop6) | |
| Conv7 | 3 | 128/64 | MaxPool(conv3+Fop2)+up7 | |
| Fop7 | | 128/64 | MaxPool(conv3+Fop2)+up7 | 8 |
| up8 | 2 | 64/32 | UpSample(conv7+Fop7) | |
| Conv8 | 3 | 64/32 | MaxPool(conv3+Fop2)+up8 | |
| Fop8 | | 64/32 | MaxPool(conv3+Fop2)+up8 | 16 |
| up9 | 2 | 32/32 | UpSample(conv8+Fop8) | |
| Conv9 | 3 | 32/32 | conv1+up9+inputs | |
| Fop9 | | 32/32 | conv1+up9+inputs | 32 |
| residual | 1 | 32/3 | conv9+Fop9 | |

As in StegaStamp(Tancik et al., 2020) and Hidden(Zhu et al., 2018), if we divide $C(p)$ by the number of pixels from the original image, we find the number of bits-per-pixel encoded by this method.

## A.3 PRINTED IMAGES

In this section, we present examples of encoded images generated by FStega. All these images were printed on A4 paper sheets using two office printers: a Brother L3270CDW and an Epson ET8500. The photos were captured using a Samsung S22 Plus mobile phone camera. As previously demonstrated in our experiments, we printed 100 images for each image size (10x10cm and 15x15cm) and took 10 photographs of each printed image. In total, we tested 2,000 captured images, with our decoder successfully decoding each image. Figures 7 and 8 show examples of the printed encoded images.

Table 5: We identify convolutional layers with the prefix "Conv", the visual transformer as "Vit", Fourier Operators as Fop, and Fc as a fully connected layer. A GELU is applied after each layer except the last. For Fc2, we apply a Sigmoid activation function.

| Layer | Kernel | ch | Input | Modes | Strides |
|---|---|---|---|---|---|
| Input | | 3 | encoded_image | | |
| Conv1 | 3 | 3/32 | Input | | 2 |
| Fop1 | | 32/32 | Conv1 | 32 | |
| Conv2 | 3 | 32/32 | Fop1 | | 2 |
| Conv3 | 3 | 32/64 | Conv2 | | 4 |
| Fop2 | | 32/64 | Conv3 | 16 | |
| Conv4 | 3 | 64/64 | Fop2 | | 4 |
| Conv5 | 3 | 64/128 | Conv4 | | 8 |
| Vit | | 128/128 | Conv5 | | |
| fc1 | | 40352/1024 | Flatten(Vit) | | |
| fc2 | | 1024/100 | Dense1 | | |

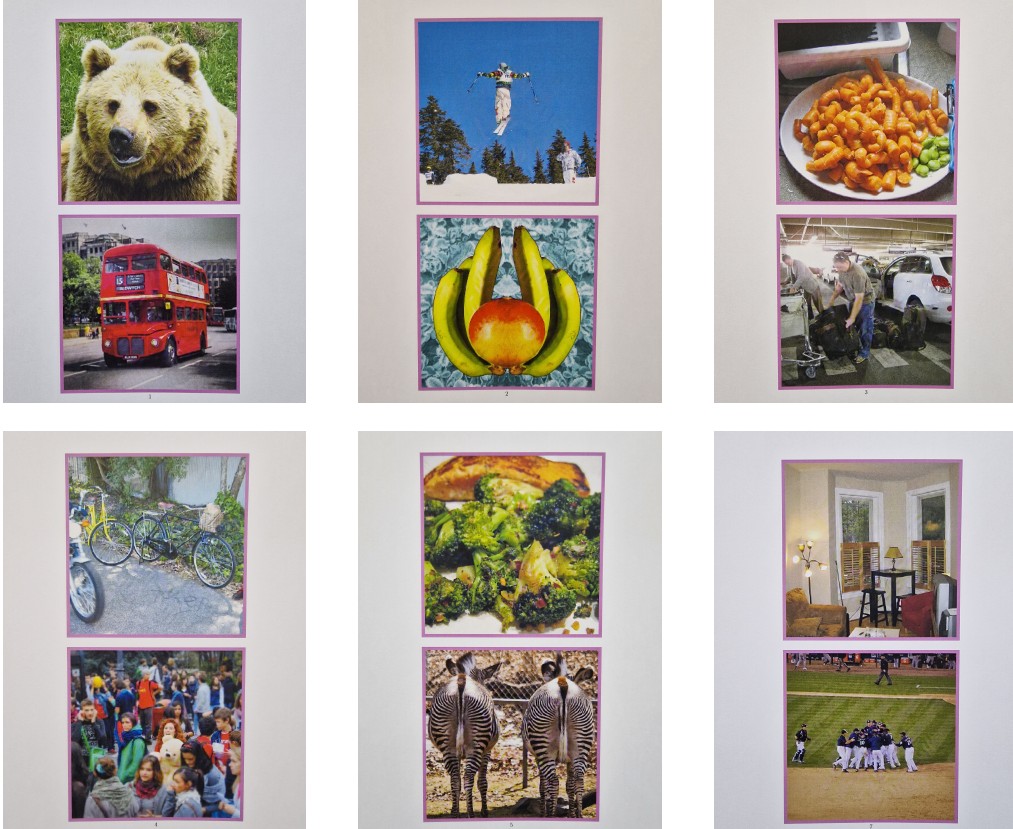

Figure 7: Printed images with dimesion of 10x10cm.All printed images are part of the test set of the COCO dataset.

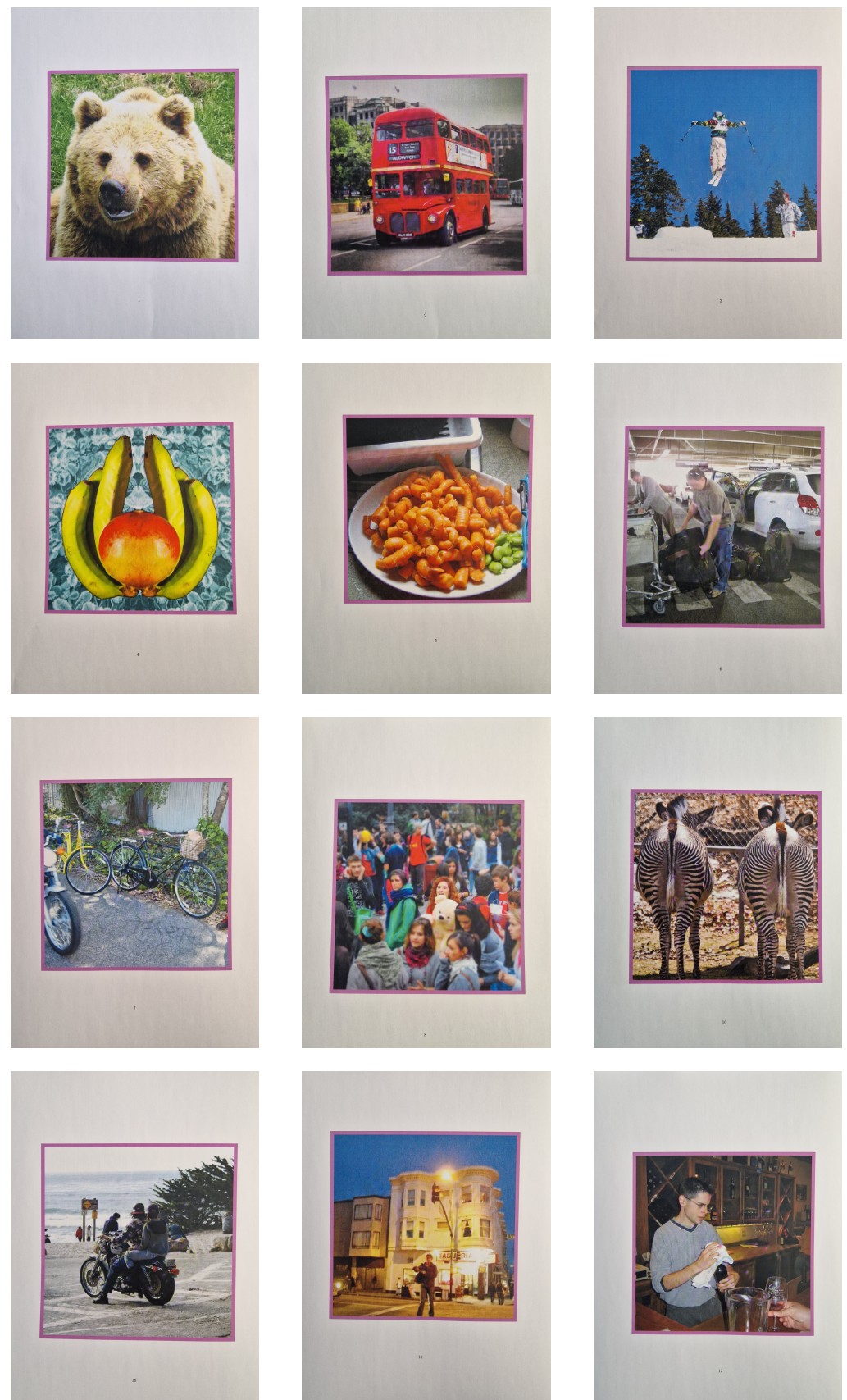

Figure 8: Printed images with dimesion of 15x15cm.All printed images are part of the test set of the COCO dataset.

