# OpenReview forum: "FStega: Fourier Neural Operators for printer-proof steganography"
_ICLR.cc/2024/Conference — ICLR 2024 Conference Withdrawn Submission_

### Official Review · Reviewer_JGfL · 2023-10-29

**Soundness:** 2 fair
**Presentation:** 2 fair
**Contribution:** 1 poor
**Rating:** 1
**Confidence:** 5

**Summary:**

The paper proposes steganography which is robust against print & scan attack. The capacity of the model is low, since it embeds 100 bits into 256x256x3 image, which makes it more akin watermarking then steganography. There is no proper evaluation of the security of steganographic method (see for example [1]), which again makes the work more akin to watermarking. Continuing on this, examples in Figure 5 shows a visible distortion of images, which will make the steganography likely very detectable. The paper completely ignores the prior art method which are not based on deep networks (see for example [2] and papers citing this work) and therefore fails to compare with it. As such, it is really difficult to judge the advancement over the prior art.

[1] Bernard, Solène, et al. "Backpack: a Backpropagable Adversarial Embedding Scheme." IEEE Transactions on Information Forensics and Security 17 (2022): 3539-3554.

[2] Solanki, Kaushal, et al. "Print and scan'resilient data hiding in images." IEEE Transactions on Information Forensics and Security 1.4 (2006): 464-478.

**Strengths:**

The model of the print and scan process is fair

**Weaknesses:**

* It seems to me the paper describes more watermarking rather than steganography, as the security of the scheme is not evaluated and the capacity of the scheme is very low.
* The paper ignores prior art not based on deep neural networks, which I think is not good practice in sound science.

**Questions:**

No questions

---

### Official Review · Reviewer_owcE · 2023-10-31

**Soundness:** 2 fair
**Presentation:** 2 fair
**Contribution:** 1 poor
**Rating:** 3
**Confidence:** 4

**Summary:**

This paper proposes a data hiding scheme by incorperating the Fourier Neural Operator for the design of secret encoder. It also simulates the print-recapture attack layers to improve the robustness of the scheme.

**Strengths:**

1. The use of Fourier Neural Operator for the design of secret encoder is shown to be able to improve the decoding accuracy.
2. The simulation of print-recapture processing to improve the robustness.

**Weaknesses:**

1. Concept misuse. The purpose of steganography is for convert communication, which requires the stego-image to be indistinguishable from the original image visually and statistically. This paper only considers the visual quality of the stego-image, the statistial imperceptibility (i.e., the undetectability) of the stego-image is not evaluated at all. As a matter of fact, this is a robust watermarking scheme rather than a steganographic scheme.

2. Limited technical contribution. Putting the misuse of concept aside, the technical contribution if rather limited. The both the network structure and the operator are existing schemes. It is also nothing new to add a attack layer to improve the robustness, even though the similation layer itself might be different from the existing simulation layers.

3. Incremental improvement. According to table 3, the improvement of the proposed scheme over the SOTA is rather limited.

**Questions:**

see weakness

---

### Official Review · Reviewer_pVko · 2023-11-04

**Soundness:** 2 fair
**Presentation:** 2 fair
**Contribution:** 2 fair
**Rating:** 6
**Confidence:** 3

**Summary:**

1. The authros praposed  a method based on Fourier Neural Operator to embed bitstrings in images. The method is able to learn critical frequencies from the image and the message to improve the decoding process of the hidden data.
2. Robust decoding method.

**Strengths:**

1.Novelity of the paper is good.

**Weaknesses:**

1.Related work should be discussed in detail.
2. More detailed analysis of results need to be presented.
2.Mention the experimental set up.

**Questions:**

1.Related work should be discussed in detail.
2. More detailed analysis of results need to be presented.
2.Mention the experimental set up.

---

### Official Review · Reviewer_E8yC · 2023-11-04

**Soundness:** 2 fair
**Presentation:** 2 fair
**Contribution:** 1 poor
**Rating:** 3
**Confidence:** 2

**Summary:**

The paper addresses the challenge of hiding and extracting messages in printed images while balancing data recovery accuracy and visual quality. It introduces a method using Fourier Neural Operator to embed bitstrings in images, leveraging critical frequencies from the image and message to enhance the decoding process. The paper also outlines an improved noise simulation process and a robust decoder, incorporating convolutional layers and a vision transformer to mitigate noise effects when images are printed and captured. Experimental results verified the method's ability to accurately retrieve hidden data from various printed images under different lighting and perspective conditions, achieving 100% accuracy in recovery for two different image sizes.

**Strengths:**

To the best of my efforts, the key strength identified is the introduction of the Fourier Neural Operator, which could guide the data embedding in the texture regions.

**Weaknesses:**

1) Related works. The background context and literature review are somewhat insufficient. While the paper addresses the crucial issue of balancing data recovery accuracy and visual quality, it would greatly benefit from a more comprehensive discussion of the existing methods and challenges in this domain. I recommend the authors provide a more thorough analysis of the prior research and related techniques in the printer-proof robust watermarking field. This will help contextualize their work within the broader research landscape and underscore the novelty and significance of their contributions.
2) Novelty. The manuscript offers valuable insights into the application of existing methods, such as the Fourier Neural Operator, for data embedding in printed images. However, I must highlight the lack of substantial innovation in the current work, which primarily appears to be more of an engineering practice. The paper would greatly benefit from a more profound exploration of the critical challenges inherent in printer-proof watermarking. For instance, the authors could delve into discussing crucial aspects like the localization of the embedded regions and the development of techniques enabling watermark extraction from incomplete watermarked image regions. In addition, the performance of the proposed FStega is comparable to the method RoSteALS, and sometimes slightly inferior to RoSteALS. A more comprehensive analysis of these fundamental issues would significantly contribute to the novelty and scholarly impact of the research.
3) Contribution. The authors should clearly summarize the contribution of the proposed method.
4) Limitation. The proposed method is not robust to the image scaling operation. In fact, there exist works emphasizing this issue, e.g., [C1].
[C1] Qingliang Liu, Jiangqun Ni, and Xianglei Hu. 2023. Robust Image Steganography against General Scaling Attacks. In Proceedings of the 31st ACM International Conference on Multimedia (MM '23). Association for Computing Machinery, New York, NY, USA, 8233–8241. https://doi.org/10.1145/3581783.3612267

**Questions:**

2) Novelty:
- How does the proposed method utilizing the Fourier Neural Operator offer a significant advancement in the context of printer-proof watermarking?
- What are the precise differences in performance between the proposed FStega and the RoSteALS method, and how do these findings contribute to the understanding of the proposed method's novelty and significance?

3) Contribution:
- What is the unique contribution of the proposed method compared to existing techniques in printer-proof watermarking?
- What specific advancements or improvements does the proposed method offer in the realm of printer-proof robust watermarking?

4) Limitations:
- How does the proposed method currently handle the image scaling operation, and how does it compare to the approaches highlighted in the referenced work by work [C1]?
- What potential modifications or enhancements could be suggested to improve the robustness of the proposed method against image scaling operations?

[C1] Qingliang Liu, Jiangqun Ni, and Xianglei Hu. 2023. Robust Image Steganography against General Scaling Attacks. In Proceedings of the 31st ACM International Conference on Multimedia (MM '23). Association for Computing Machinery, New York, NY, USA, 8233–8241. https://doi.org/10.1145/3581783.3612267

---

### Official Review · Reviewer_3bbC · 2023-11-07

**Soundness:** 2 fair
**Presentation:** 3 good
**Contribution:** 2 fair
**Rating:** 1
**Confidence:** 5

**Summary:**

This paper proposed a new printer-proof steganography method. This method uses Fourier Neural Operator to embed secret messages into images. Besides, this paper employs an noise-simulation layer to improve its robustness against printing and scanning.

**Strengths:**

- This paper is well orgnized.
- This paper provides a comprehensive analysis of its limitations.
- This paper pays attention to printing simulation and camera noise, which is rarely concerned by previous papers

**Weaknesses:**

- In Introduction, this paper list three limitations of Stegastamp. However, the second point is untenable. The author says that "it (Stegastamp) has difficulty in decoding images with sizes smaller than 15 cm width". First, is this statement true? I suppose not. I used to print StegaStamp encoded images at 6cm width. But the decode accuracy still remains 98%. I don't think this can be called "having difficulty". Second, the width of the printed image is both determined by printing resolution and image size (pixel). Just using the image width is not convincing enough.
- Related work is inadequately introduced.

- Several details need verification or enhancement, including:
a. Lack of training specifics and the absence of specific data when discussing network size.
b. Lack of alignment between sections, as L2 Loss is mentioned in the Introduction but not in Section 5.2.
c. Numerous references to data results that are not presented in tables, for example, Table 3 should include a column for method accuracy.
d. Typo errors, such as ',Light field messaging' in Section 6.2.

- The experiments in this paper are insufficient, unconvincing, and unreasonable.
  + INSUFFICIENT. The amount of experiment is too small. This paper only presents 2 quantitative experiments (Table 2 and Table 3). This paper adopts a complex noise simulation layer to simulate real-life distortions. However, there were no relevant experiments (for example, shooting at different distances or angles to test the performance under warping).
  + UNCONVINCING. Among the two experiments in this paper, the first (Table 2) is performed on printed images. The other (Table 3) is conducted in digital environment. However, isn’t the core purpose of this paper exactly to solve the problem of watermark extraction in the printing scenario? How could experiment be convincing without conduting on printed images?
  + UNREASONABLE. This is a continuation of the previous point. The setting of the experiment in Table 3 in this paper is completely unreasonable.Since this experiment adopts the noise simulation in FStega for printing simulation, the comparison on extraction accuracy between methods like StageStamp, RoSteALS, and FStega in Section 6.2 is unfair. These different methods employ varying noise simulation modules, and using the noise module in FStega (seen in training) is not well-justified.

**Questions:**

- Could the authors provide an explanation of the term 'bits-per-megapixel'? FStega and StegaStamp seem to have an embedding capacity of 100 bits, and StegaStamp can also support 200 bits, but there doesn't appear to be corresponding data for FStega.

- In the section where the Fourier operator is introduced, the authors mention the linear transform W. Could the authors provide a more detailed explanation of its specific role and perhaps offer some explanatory analysis?

- RoSteALS was not designed with printed images in mind, but it seems to be resistant to the noise simulation module used in this paper (accuracy: 99%). I suggest the authors conducting an experiment to test RoSteALS's extraction accuracy on real printed images.

- In Section 6.3, it is mentioned that FStega is sensitive to scaling operations. Why can't scaling operations be introduced in the noise module?